# Spectra of a Rydberg Atom in Crossed Electric and Magnetic Fields

**Andrei Letunov [1] and Valery Lisitsa [1,2,*]**

[1]    National Research Centre "Kurchatov Institute", 123182 Moscow, Russia; letunovandrey11@yandex.ru
[2]    National Research Nuclear University MEPhI, 115409 Moscow, Russia
*    Correspondence: vlisitsa@yandex.ru

**Abstract:** Contemporary spectroscopic studies of astrophysical and laboratory plasmas frequently deal with extremely large values of principle quantum numbers of atomic systems. These atomic states are very sensitive to electric and magnetic fields of the surrounding medium. While interpreting the spectra of such excited atomic systems, one faces the problem of a huge array of radiative transitions between highly excited atomic levels. Moreover, external electric and magnetic fields significantly complicate the problem because of the absence of standard selection rules typical for the spherical quantization. The analytical expression in the parabolic representation for dipole matrix elements obtained by Gordon contains hyper-geometric series and it has a very complex structure. The matrix elements that involve the presence of electric and magnetic fields are calculated while using a representation closely related to the parabolic quantization on two different axes. This matrix element depends in a complex way on the transition probabilities in the parabolic coordinate system (Gordon's formulas) and the Wigner d-functions. This circumstance leads to even greater computational difficulties. A method of simplification of these complicated expressions for transition probabilities is demonstrated. The semiclassical approximation for coordinate matrix elements (Gulayev) and recurrence properties of the Wigner d-functions are used. The $H_{n\beta}$ line is under consideration. Specific calculations for the transition 10–8 in the case of parallel and perpendicular fields are presented.

**Keywords:** atomic physics; rydberg atoms; stark-zeeman effect

## 1. Introduction

Investigation of spectra of a highly excited (Rydberg) hydrogen is an important tool for studying the physical properties of the H II regions and the interstellar medium. The application of Rydberg spectral lines to astrophysical problems is presented in [1–8]. Often, while studying astrophysical plasmas, one has to deal with highly excited atomic levels [4–8]. However, one faces two fundamental problems. The first one is connected with the influence of external electric **F** and magnetic **B** fields on spectra of Rydberg atoms. This problem is related to the combined Stark–Zeeman effect. It turns out that a suitable description of a hydrogen atom in external **F** − **B** fields requires a transition to a special basis that is associated with taking the symmetry properties of the Coulomb field into account. The second problem is related to the complicated structure of the array of radiative transitions between Rydberg atomic states. Hence, it seems to be a very complicated problem to find a reasonable treatment for the array of spectral lines transition probabilities in the parabolic quantum numbers presentation with respect to the adequate description of the array. In the present work, we show how one can obtain universal formulas for the radiation intensity of a hydrogen-like atom in external electric and magnetic fields.

In order to describe Stark broadening in plasmas, it is convenient to use the parabolic representation, instead of the spherical coordinate system. However, the expressions for dipole matrix elements in this

basis obtained by Gordon [9] contain hyper-geometric series, which makes calculations of intensities very cumbersome. Moreover, one faces a huge growth of the transition array for Rydberg atomic states (it grows proportionally to $n^4$, where n is the principle quantum number). A solution to this problem was given by Gulayev in [10,11]. He obtained the semiclassical approximation for coordinate matrix elements. The problem of the joint action of crossed electric and magnetic fields on Rydberg atomic states still is not solved properly—in the sense of making it possible to calculate such spectra.

Transition probabilities in the spherical coordinate system have been deeply studied in different limits [12–14]. Additionally, the orbital quantum number l follows the selection rule, which allows for one to make fast calculations of dipole matrix elements. However, the energy shift in a constant electric field has a simple form in the parabolic representation. In the present paper, the problem of a large transition array will be solved for highly exited energy levels by establishing approximate selection rules for parabolic quantum numbers. For the first time a hydrogen atom in external electric **F** and magnetic **B** fields was considered in the framework of classical mechanics in [15]. The quantum treatment was presented in [16]. The symmetry of the Coulomb field can be used to change the representation. The Hamiltonian of the electron in the Coulomb field and external **F-B** fields has the following form

$$H = \frac{\mathbf{p}^2}{2} - \frac{Z}{r} + \mathbf{Fr} + \frac{1}{2c}\mathbf{BL} \tag{1}$$

Here, **p**,**r** and **L** are the momentum,the coordinate and the angular momentum operators of the electron, respectively, Z is the nuclear charge. This Formula (1) and every other in this paper is written in the atomic units. The perturbation $\mathbf{Fr} + \frac{1}{2c}\mathbf{BL}$ can be rewritten in another way.

$$\Delta H = \mathbf{Fr} + \frac{1}{2c}\mathbf{BL} = E_1\mathbf{J_1} + E_2\mathbf{J_2} \tag{2}$$

where

$$\mathbf{J_{1,2}} = \frac{1}{2}(\mathbf{L} \pm \mathbf{A}) \tag{3}$$

**A** is the specific constant of motion in the Coulomb field—the Runge–Lenz vector.

The O(4) symmetry of the hydrogen atom allows one to change the representation to the quantum numbers related to the projections of vectors (3) on arbitrary axis. The additional level of the symmetry in the Coulomb field is connected with the conservation of the Runge–Lenz vector.

$$\mathbf{E}_{1,2} = \frac{1}{2c}\mathbf{B} \mp \frac{3}{2}n\mathbf{F} \tag{4}$$

We can do this, because in the Coulomb field there is a relation between the Runge–Lenz vector and the radius-vector:

$$\mathbf{A} = -\frac{2}{3n}\mathbf{r} \tag{5}$$

The energy shift is equal to

$$\Delta E = E_1 n' + E_2 n'' \tag{6}$$

where $n'$ and $n''$ are projections of (3) on vectors (4).

The vectors (3) have properties of an angular momentum(see for example [17]). Moreover, projections of (3) on the same direction (z-axis) are related to the parabolic quantum numbers [17]

$$\begin{cases} i_2 - i_1 = n_1 - n_2 \\ i_2 + i_1 = m \end{cases} \tag{7}$$

where $i_{1,2}$ are projections of (3) on z direction(quantization axis) and m is the magnetic quantum number.

Using the angular momentum properties of (3), one can change the representation from $i_1, i_2$ to $n', n''$.

$$|n, n', n''> = \sum_{i_1=-j}^{j} \sum_{i_2=-j}^{j} d^j_{i_1 n'}(\alpha_1) d^j_{i_2 n''}(\alpha_2) |n, i_1, i_2 > \qquad (8)$$

where $d^j_{m_1 m_2}(\beta)$ is the Wigner d-function.

$$j = \frac{n-1}{2} \qquad (9)$$

Here, in (8), $\alpha_{1,2}$ are the angles between vectors $\mathbf{J_{1,2}}$ and $\mathbf{E_{1,2}}$.

$$cos\alpha_{1,2} = \frac{\frac{1}{2c}B \mp \frac{3}{2}nFcos\theta}{E_{1,2}} \qquad (10)$$

where $\theta$ is the angle between the electric **F** and magnetic fields **B**.

We want to underline the fact that the additional degeneracy can be obtained in the case of perpendicular crossed electric and magnetic fields. Particularly, the energy shift depends on $n' + n''$ when $\theta = \frac{\pi}{2}$. One can find a detailed consideration of this case in [18].

A calculation of dipole matrix elements in this representation was presented in [19]. The general expression for coordinate matrix element in the basis of $n', n''$ has the following form

$$a^{\bar{n} \bar{n}' \bar{n}''}_{nn'n''} = \sum_{\bar{i}_1=-\bar{j}}^{\bar{j}} \sum_{\bar{i}_2=-\bar{j}}^{\bar{j}} \sum_{i_1=-j}^{j} \sum_{i_2=-j}^{j} d^{\bar{j}}_{\bar{i}_1 \bar{n}'}(\bar{\alpha}_1) d^{\bar{j}}_{\bar{i}_2 \bar{n}''}(\bar{\alpha}_2) d^{j}_{i_1 n'}(\alpha_1) d^{j}_{i_2 n''}(\alpha_2) a^{\bar{n} \bar{i}_1 \bar{i}_2}_{n i_1 i_2} \qquad (11)$$

where $a = X, Y, Z$. Here n relates to the upper atomic state.

As a result, the number of terms in (11) grows proportionally to $n^4$. However, the use of Gulayev's results and the specific properties of the Wigner d-functions allows for one to make a significant simplification of (11). In the present paper, we consider the $H_{n\beta}$ ($\Delta n = n - \bar{n} = 2$) lines.

## 2. Derivation of Dipole Matrix Elements

Our purpose is a simplification of the complicated Formula (11). The main problem with this expression is the presence of four sums. The number of terms in this sum is proportional to $n^4$. The Wigner d-functions also have a complex structure. They can be analytically expressed in terms of the Jacobi polynomials [20]. The main idea is to use the combination of the important results from [10,11] and the d-function properties [20].

In works [10,11] the authors introduced a new quantum number K

$$K = (n_1 - n_2) - (\bar{n}_1 - \bar{n}_2) \qquad (12)$$

The energy shift can be rewritten by using the quantum number K as follows

$$\frac{\Delta E}{\omega_F} = Kn + \Delta nk \qquad (13)$$

where $k = \bar{n}_1 - \bar{n}_2$ and $\omega_F = \frac{3}{2}F$

It turns out that the intensity of the radiation in the dipole approximation strongly depends on the number K from (12). The transitions with some specific values of K make a greater contribution to the intensity of the radiation than the others. As for the $H_{n\beta}$, lines one needs to calculate transitions only with $K = \pm 1$ and $K = \pm 2$.

$$Z^m_m = \frac{1}{4}b\left[(\bar{n}_1 + m + 2)(\bar{n}_1 + 2)\delta_{K,+2} + (\bar{n}_2 + m + 2)(\bar{n}_2 + 2)\delta_{K,-2}\right] \qquad (14)$$

where $b^2 = \frac{4n\bar{n}}{(n-\bar{n})^2}$. Here, m is the absolute value of the magnetic quantum number and $\delta_{k,l}$ is the Kronecker delta symbol.

We can split transitions into special series corresponding to specific values of K. Thus, we will obtain two selection rules: the first one for the K-determination of series, the second one is the selection rule for magnetic quantum number m. Proceeding to the $i_1, i_2$ representation, we obtain approximate selection rules for the parabolic quantum numbers.

$$\begin{cases} (i_2 - i_1) - (\bar{i}_2 - \bar{i}_1) = \pm 2 \\ i_2 + i_1 = \bar{i}_2 + \bar{i}_1 \end{cases} \tag{15}$$

The solution of the system (15) has the following form

$$\begin{cases} i_2 = \bar{i}_2 \pm 1 \\ i_1 = \bar{i}_1 \mp 1 \end{cases} \tag{16}$$

The parabolic quantum numbers satisfy the following relation

$$n = n_1 + n_2 + |m| + 1 \tag{17}$$

Using (17) and (7), one can obtain the relation between $n_1, n_2$ and $i_1, i_2$

$$\begin{cases} n_1 = \frac{n - |i_1 + i_2| + i_2 - i_1 - 1}{2} \\ n_2 = \frac{n - |i_1 + i_2| + i_1 - i_2 - 1}{2} \end{cases} \tag{18}$$

Subsequently, it is necessary to substitute (14), (16) and (18) into (11).

$$Z_{1,2nn'n''}^{\bar{n}\bar{n}'\bar{n}''} = \sum_{\bar{i}_1=-\bar{j}}^{\bar{j}} \sum_{\bar{i}_2=-\bar{j}}^{\bar{j}} d_{\bar{i}_1\bar{n}'}^{\bar{j}}(\alpha_1) d_{\bar{i}_2\bar{n}''}^{\bar{j}}(\alpha_2) d_{\bar{i}_1\pm 1 n'}^{j}(\alpha_1) d_{\bar{i}_2 \mp 1 n''}^{j}(\alpha_2) G_{1,2}(\bar{i}_1, \bar{i}_2) \tag{19}$$

where

$$G_1 = (\frac{n}{2} - \bar{i}_1)(\frac{n}{2} + \bar{i}_2)$$

$$G_2 = (\frac{n}{2} + \bar{i}_1)(\frac{n}{2} - \bar{i}_2)$$

After that, we have to use recurrence relations for the d-functions [20]

$$d_{m_1,m_2}^{j}(\beta) = \sqrt{\frac{j - m_2}{j - m_1}} \cos(\frac{\beta}{2}) d_{m_1+\frac{1}{2},m_2+\frac{1}{2}}^{j-\frac{1}{2}}(\beta) - \sqrt{\frac{j + m_2}{j - m_1}} \sin(\frac{\beta}{2}) d_{m_1+\frac{1}{2},m_2-\frac{1}{2}}^{j-\frac{1}{2}}(\beta) \tag{20}$$

$$d_{m_1,m_2}^{j}(\beta) = \sqrt{\frac{j - m_2}{j - m_1}} \sin(\frac{\beta}{2}) d_{m_1-\frac{1}{2},m_2+\frac{1}{2}}^{j-\frac{1}{2}}(\beta) + \sqrt{\frac{j + m_2}{j - m_1}} \cos(\frac{\beta}{2}) d_{m_1-\frac{1}{2},m_2-\frac{1}{2}}^{j-\frac{1}{2}}(\beta) \tag{21}$$

In order to use (20) and (21), it is necessary to use these relations twice: for $Z_{1nn'n''}^{\bar{n}\bar{n}'\bar{n}''}$ relation (20), for $Z_{2nn'n''}^{\bar{n}\bar{n}'\bar{n}''}$ (21). In the limit $n, \bar{n} \gg 1$, one can notice that factors in the recurrence relations and $G_{1,2}$ coincide. It allows for one to use the orthogonality relation for d-functions

$$\sum_{m_3=-j}^{j} (-1)^{m_3-m_2} d_{m_2,m_3}^{j}(\beta) d_{m_3,m_1}^{j}(\beta) = \delta_{m_1,m_2} \tag{22}$$

After that, we can obtain the Z-coordinate matrix element in the representation of states (8)

$$Z^{\bar{n}\bar{n}'\bar{n}''}_{nn'n''} = \frac{1}{4}b(-1)^{\Delta\bar{n}'+\Delta\bar{n}''}\left[Z^{\bar{n}\bar{n}'\bar{n}''}_{1nn'n''} - Z^{\bar{n},\bar{n}',\bar{n}''}_{2nn'n''}\right] \tag{23}$$

$$Z^{\bar{n}\bar{n}'\bar{n}''}_{1nn'n''} = \left[(\frac{n}{2}-n')cos^2\left(\frac{\alpha_1}{2}\right)\delta_{\bar{n}',n'+1} - sin(\alpha_1)\sqrt{(\frac{n}{2}-n')(\frac{n}{2}+n')}\delta_{\bar{n}',n'} + \right.$$
$$+(\frac{n}{2}+n')sin^2\left(\frac{\alpha_1}{2}\right)\delta_{\bar{n}',n'-1}\right] \times \left[(\frac{n}{2}-n'')sin^2\left(\frac{\alpha_2}{2}\right)\delta_{\bar{n}'',n''+1} + \right.$$
$$+sin(\alpha_2)\sqrt{(\frac{n}{2}-n'')(\frac{n}{2}+n'')}\delta_{\bar{n}'',n''} + (\frac{n}{2}+n'')cos^2\left(\frac{\alpha_2}{2}\right)\delta_{\bar{n}'',n''-1}\right]$$

$$Z^{\bar{n}\bar{n}'\bar{n}''}_{2nn'n''} = \left[(\frac{n}{2}-n'')cos^2\left(\frac{\alpha_2}{2}\right)\delta_{\bar{n}'',n''+1} - sin(\alpha_2)\sqrt{(\frac{n}{2}-n'')(\frac{n}{2}+n'')}\delta_{\bar{n}'',n''} + \right.$$
$$+(\frac{n}{2}+n'')sin^2\left(\frac{\alpha_2}{2}\right)\delta_{\bar{n}'',n''-1}\right] \times \left[(\frac{n}{2}-n')sin^2\left(\frac{\alpha_1}{2}\right)\delta_{\bar{n}',n'+1} + \right.$$
$$+sin(\alpha_1)\sqrt{(\frac{n}{2}-n')(\frac{n}{2}+n')}\delta_{\bar{n}',n'} + (\frac{n}{2}+n'')cos^2\left(\frac{\alpha_1}{2}\right)\delta_{\bar{n}',n'-1}\right]$$

Derivation of the expression for the X-matrix element is similar to the Z-case. The selection rules for K and for the magnetic quantum number are described by the following system

$$\begin{cases} (i_2 - i_1) - (\bar{i}_2 - \bar{i}_1) = \pm1 \\ |i_2 + i_1| = |\bar{i}_2 + \bar{i}_1| \pm 1 \end{cases} \tag{24}$$

System (24) leads to four possibilities

$$\begin{cases} \begin{cases} i_2 = \bar{i}_2 \pm 1 \\ i_1 = \bar{i}_1 \end{cases} & \begin{cases} i_2 = \bar{i}_2 \\ i_1 = \bar{i}_1 \pm 1 \end{cases} \end{cases} \tag{25}$$

Expressions for the X-matrix elements correspond to $K \pm 1$

$$X^{m-1}_m = \frac{1}{4}b\left[\sqrt{n_1(n_1+m)(\bar{n}_1+m)(\bar{n}_2+m)}\delta_{K,+1} + \sqrt{n_2(n_2+m)(\bar{n}_2+m)(\bar{n}_2+m)}\delta_{K,-1}\right] \tag{26}$$

$$X^{m+1}_m = \frac{1}{4}b\left[\sqrt{n_1(n_1+m)\bar{n}_1\bar{n}_2}\delta_{K,+1} + \sqrt{n_2(n_2+m)\bar{n}_2\bar{n}_2}\delta_{K,-1}\right] \tag{27}$$

In the case of the X-dipole matrix element, one can obtain the expression that is similar to (19)

$$X^{\bar{n}\bar{n}'\bar{n}''}_{1,2,3,4nn'n''} = \sum_{\bar{i}_1=-\bar{j}}^{\bar{j}}\sum_{\bar{i}_2=-\bar{j}}^{\bar{j}} d^{\bar{j}}_{\bar{i}_1\bar{n}'}(\alpha_1)d^{\bar{j}}_{\bar{i}_2\bar{n}''}(\alpha_2)d^{j}_{\bar{i}_1\pm1n'}(\alpha_1)d^{j}_{\bar{i}_2\mp1n''}(\alpha_2)F_{1,2,3,4}(\bar{i}_1,\bar{i}_2) \tag{28}$$

where

$$F_1 = (\frac{n}{2}+\bar{i}_2)\sqrt{(\frac{n}{2}-i_1)(\frac{n}{2}+i_2)}$$

$$F_2 = (\frac{n}{2}+\bar{i}_1)\sqrt{(\frac{n}{2}+i_1)(\frac{n}{2}-i_2)}$$

$$F_3 = (\frac{n}{2}-\bar{i}_2)\sqrt{(\frac{n}{2}+i_1)(\frac{n}{2}-i_2)}$$

$$F_4 = \left(\frac{n}{2} - \bar{i}_1\right)\sqrt{\left(\frac{n}{2} - i_1\right)\left(\frac{n}{2} + i_2\right)}$$

In order to achive the coincidence of mutual factors in the recurrence relations and expressions ((26) and (27)), one has to use ((20) and (21)), in a special way. For the first case in (25), it is necessary to use relation (20) and, after that, relation (21). In distinction, for the second case in (25) one has to use (20) after (21). After these manipulations, it is easy to obtain the X-matrix element

$$X_{nn'n''}^{\bar{n}\bar{n}'\bar{n}''} = \frac{1}{4}b(-1)^{\Delta\bar{n}'+\Delta\bar{n}''}\left[X_{1nn'n''}^{\bar{n}\bar{n}'\bar{n}''} - X_{2nn'n''}^{\bar{n},\bar{n}',\bar{n}''} - X_{3nn'n''}^{\bar{n}\bar{n}'\bar{n}''} + X_{4nn'n''}^{\bar{n}\bar{n}'\bar{n}''}\right] \tag{29}$$

$$X_{1nn'n''}^{\bar{n}\bar{n}'\bar{n}''} = \left[\left(\frac{n}{2} - n'\right)\cos^2\left(\frac{\alpha_1}{2}\right)\delta_{\bar{n}',n'+1} - \sin(\alpha_1)\sqrt{\left(\frac{n}{2} - n'\right)\left(\frac{n}{2} + n'\right)}\delta_{\bar{n}',n'} + \right.$$
$$\left. +\left(\frac{n}{2} + n'\right)\sin^2\left(\frac{\alpha_1}{2}\right)\delta_{\bar{n}',n'-1}\right] \times \left[\frac{1}{2}\sin(\alpha_2)\left(\left(\frac{n}{2} - n''\right)\delta_{\bar{n}'',n''+1} - \left(\frac{n}{2} + n''\right)\delta_{\bar{n}'',n''-1}\right) + \right.$$
$$\left. +\delta_{\bar{n}'',n''}\cos(\alpha_2)\sqrt{\left(\frac{n}{2} + n''\right)\left(\frac{n}{2} - n''\right)}\right]$$

$$X_{2nn'n''}^{\bar{n}\bar{n}'\bar{n}''} = \left[\left(\frac{n}{2} - n''\right)\sin^2\left(\frac{\alpha_2}{2}\right)\delta_{\bar{n}'',n''+1} + \sin(\alpha_2)\sqrt{\left(\frac{n}{2} - n''\right)\left(\frac{n}{2} + n''\right)}\delta_{\bar{n}'',n''} + \right.$$
$$\left. +\left(\frac{n}{2} + n''\right)\cos^2\left(\frac{\alpha_2}{2}\right)\delta_{\bar{n}'',n''-1}\right] \times \left[\frac{1}{2}\sin(\alpha_1)\left(\left(\frac{n}{2} - n'\right)\delta_{\bar{n}',n'+1} - \left(\frac{n}{2} + n'\right)\delta_{\bar{n}',n'-1}\right) + \right.$$
$$\left. +\delta_{\bar{n}',n'}\cos(\alpha_1)\sqrt{\left(\frac{n}{2} + n'\right)\left(\frac{n}{2} - n'\right)}\right]$$

Here, $X_{3nn'n''}^{\bar{n}\bar{n}'\bar{n}''}$ can be obtained by switching $n' \Leftrightarrow n''$ (for bar values too) and $\alpha_1 \Leftrightarrow \alpha_2$ in $X_{1nn'n''}^{\bar{n}\bar{n}'\bar{n}''}$. The same connection exists between $X_{2nn'n''}^{\bar{n}\bar{n}'\bar{n}''}$ and $X_{4nn'n''}^{\bar{n}\bar{n}'\bar{n}''}$.

Using the result for the X-dipole matrix element, one can obtain the expression for the Y-matrix element. The hydrogen wave function is proportional to $e^{im\varphi}$, $X \sim \cos\varphi$, $Y \sim \sin\varphi$. Using well-known relations $\cos(z) = \frac{e^{iz}+e^{-iz}}{2}$ and $\sin(z) = \frac{e^{iz}+e^{-iz}}{2i}$, one can obtain

$$Y_{nn'n''}^{\bar{n}\bar{n}'\bar{n}''} = \frac{1}{4i}b(-1)^{\Delta\bar{n}'+\Delta\bar{n}''}\left[X_{1nn'n''}^{\bar{n}\bar{n}'\bar{n}''} - X_{2nn'n''}^{\bar{n},\bar{n}',\bar{n}''} - X_{3nn'n''}^{\bar{n}\bar{n}'\bar{n}''} - X_{4nn'n''}^{\bar{n}\bar{n}'\bar{n}''}\right] \tag{30}$$

## 3. Results

In order to analyze obtained results, we consider the ratio of Zeeman and Stark shifts denoted below as u. The intensity is proportional to the square of the absolute value of coordinate matrix element in the dipole approximation. As the intensity, we consider the function of the absolute value of dipole matrix element divided by the sum of all (X,Y,Z) intensity components.

$$u = \frac{B}{3cnE} \tag{31}$$

The reduced energy shift is equal to

$$\omega = (\bar{E}_1\bar{n}' + \bar{E}_2\bar{n}'' - E_1n' - E_2n'')/\Omega_{FB} \tag{32}$$

$$\Omega_{FB} = \frac{1}{2c}B + \frac{3}{2}nF \tag{33}$$

Figure 1 presents calculations of the intensity in the case of parallel fields. In Figure 1a, one can see pure Stark effect. This result is in agreement with [11]. The absence of the central component is typical

for the lines $H_{n\beta}$ without a magnetic field. In the presence of the magnetic field **B**, one can observe how intensity components merge together (Figure 1b,c). Finally, when the Zeeman shift becomes much larger than the Stark shift, we obtain the picture of the pure Paschen—Back effect.

Expressions (14), (26), and (27) contain Kronecker delta symbols. This circumstance leads to the fact that for highly excited levels the numbers $n', n''$ follow the selection rules.

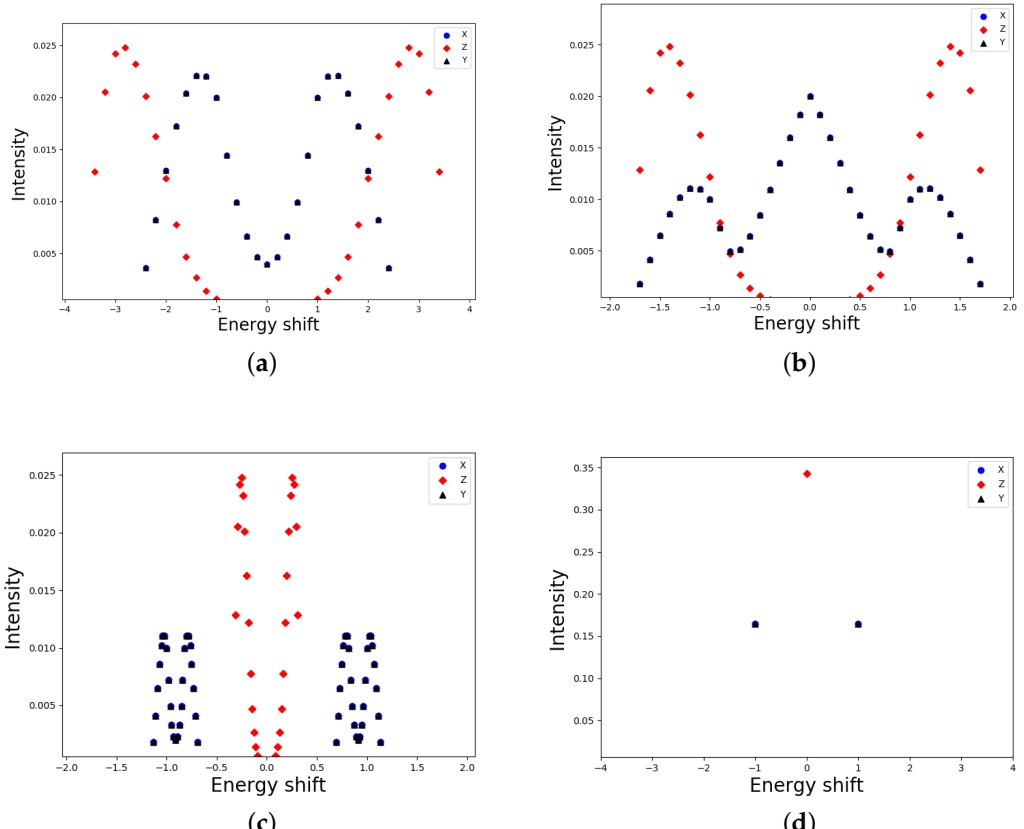

**Figure 1.** Transition from n = 10 to n = 8. Intensity (divided by the sum of intensities of all components) as the function of the reduced energy ((32) and (33)) in the case of parallel fields ($\theta = 0$): (**a**) u = 0, (**b**) u = 1 (**c**) u = 10 (**d**) u = 1000; $u = \frac{B}{3cnE}$.

In the absence of a magnetic field: $\alpha_1 = \pi$ and $\alpha_2 = 0$ (see (10)). If one substitute these values of the angles in expressions (23), (29) and(30), and change $n' \Rightarrow -n'$ one would retrieve the usual Stark effect and formulas (14), (26), and (27). In the opposite limit of the large Zeeman shift $\alpha_{1,2} = 0$ and because of the selection rules for $n', n''$ expressions (23), (29) and (30), one would reproduce the Paschen—Back effect. The intensity components of the radiation polarized in the X and Y directions coincide due to the symmetry of the system.

Figure 2a presents the case of zero magnetic field and the electric field parallel to the x-axes. It is seen that the intensity profiles, which corresponded to the X- and Z-polarizations, became interchanged. Because of the decrease in the degree of the symmetry, the matrix elements begin to appear in pairs of mismatched intensity components (Figure 2b). In comparison with the case of parallel fields, the transition to the Paschen—Back effect already occurs at $u = 10$ (Figure 2c).

The non-conservation of the full integrated intensity is related to the fact that all three types of matrix elements (polarizations) are calculated instead of two.

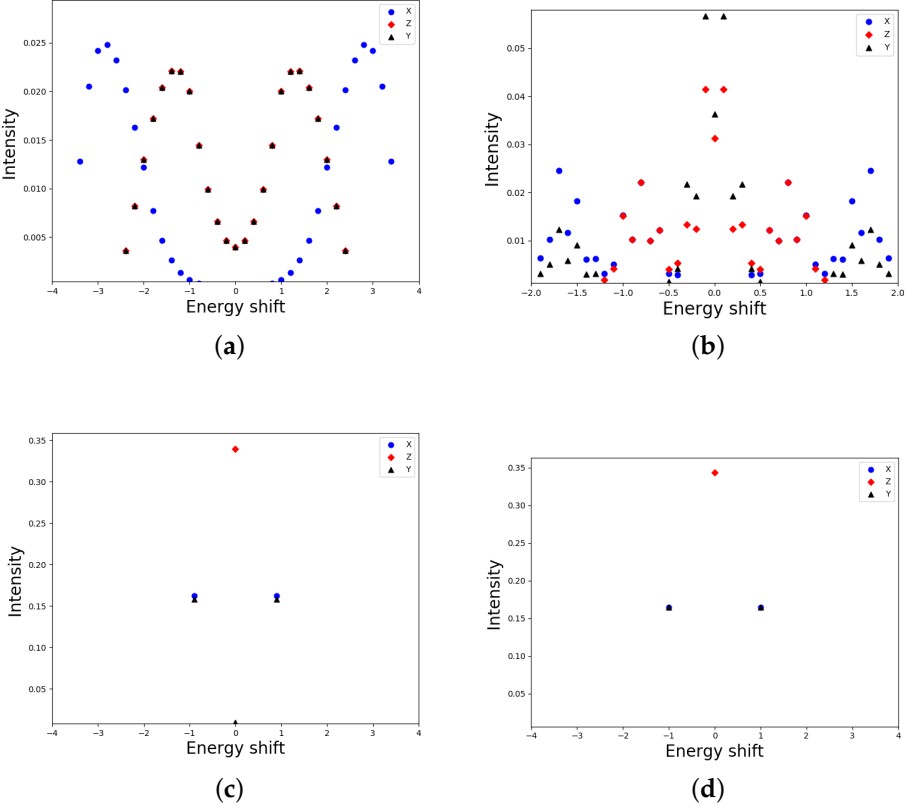

**Figure 2.** The same as in Figure 1 but for perpendicular fields ($\theta = \frac{\pi}{2}$). (**a**) u = 0, (**b**) u = 1 (**c**) u = 10 (**d**) u = 1000; $u = \frac{B}{3cnE}$.

## 4. Conclusions

Studies of astrophysical plasmas frequently employ analysis of hydrogen spectral lines. One faces two fundamental problems while dealing with highly exited (Rydberg) atomic states. The first one is related to the complex structure of accurate expressions for dipole-matrix elements in the parabolic representation obtained by Gordon [9,14]. The second one is the influence of magnetic and electric fields on spectra of a hydrogen-like atom. The solution to these problems is demonstrated in the present paper. The simultaneous use of the specific properties of the Wigner d-functions and the results that were obtained in [10,11] made it possible to simplify the complex expression (11). Practically, while dealing with large principal quantum numbers, one faces again complicated sums (11) with $n^4$ terms. It turns out that the coefficients in the recurrence relations for the Wigner d-functions coincide with the factors in the semiclassical expressions from [11]. It allows for one to use the orthogonality relation for the d-functions and get rid of four sums.

Using the semiclassical approximation for dipole matrix elements and the properties of d-functions, we reduced the complicated formula (11), which contains the complex hyper geometric series in $a_{ni_1 i_2}^{\bar{n}\bar{i}_1 \bar{i}_2}$ and the Jacobi polynomials in the d-functions, to expressions (23), (29) and(30). These formulas contain trigonometric functions and Kroneker-delta symbols, which expresses new selection rules for the quantum numbers $n'$, $n''$. Moreover, we emphasize the universality of these formulas. The new semiclassical expressions describe any transition with $\Delta n = 2$.

In Figures 1 and 2, we presented specific calculations that were related to the transition $10 - 8$. We considered the cases of parallel and perpendicular fields.Using Figures 1 and 2, it is possible to trace the smooth transition from the pure Stark effect to the Zeeman components. By gradually increasing the magnitude of the magnetic field, one can observe how the intensity components merge together.

In summary, we derived the semiclassical approximation for dipole matrix elements, while using Gulaev's formulas and the recurrence relations for the Wigner d-functions. These expressions have universal properties. The initial expression (11) contains $n^4$ terms, the d-functions, and the complicated Gordon's ([9,14]) formulas. However, the simultaneous use of both semiclassical results and the properties of d-functions leads to the simple, universal formulas (23) and (29) and (30).

**Funding:** This research received no external funding.

**Conflicts of Interest:** The authors declare no conflict of interest.

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
