# Peer review of "Spectra of a Rydberg Atom in Crossed Electric and Magnetic Fields"

_universe, doi:10.3390/universe6100157_

Round 1

Reviewer 1 Report

The manuscript

“Spectra of a Rydberg atom in crossed electric and magnetic field”

by A. Letunov et al.

proposes important simplification of matrix element calculations for Rydberg atomic states (in fact, the number of these matrix elements is proportional to the fourth power of principle quantum numbers n^4). The results demonstrate an interesting effect, namely a smooth transition from Stark to Zeeman intensities arrays.

The Rydberg atomic spectra in crossed electric and magnetic fields concerns an actual problem in atomic physics because of the computational difficulties of the transitions between highly exited atomic states. The authors considered the case of the Hn-beta spectral line as an explicit example.

The authors employ a parabolic representation that is elegant and has even partially educational aspects. One of the main problems of the parabolic representation is the absence of the selection rules for quantum numbers. Nevertheless, the authors provided a solution to this problem by establishing an approximate selection rule for Rydberg atomic states. Using the symmetry properties of the Coulomb field and the semi-classical approximation for transition probabilities a significant simplification of the general formulas for the dipole matrix elements is achieved. Moreover, the authors provide a detailed derivation for the case of the Z-axis matrix elements.

In order to increase the readership of the present publication, I would suggest to provide an Appendix, where the authors:

1) demonstrate the transformations of X matrix elements that are not presented in the manuscript

2) provide explicitly the Wigner d-coefficients to allow “easy reading” and self-contained presentation of the transformations. Here, also the recurrence relations (eqs. 20, 21) and orthogonality relation 22 could be presented and be removed from main text. Is the order of the indexes of d in eq. (11) really “i n” but not “n i” ?

Other comments:

1) the Figures need better “optical” presentation: numbers are too small, yellow color should be avoided, and units of energy explicitly states in figure captions.

2) the authors might add the Dirac notation of the matrix elements presented in eq. (11), state explicitly that nn’n’’ is the basis-set and what is upper and lower states

3) to use “i” for the projections on z-axis might be not a very convenient, when using the i-bar in indexes all formulas are difficult to read, why not to change simply to z1 and z2 and corresponding z1-bar ect. ?

4) Energy scale in Fig. 1c is not very convenient

5) The authors employ “intensity” for the ordinate of Figures, it might be useful to provide an explicit expression in the text how the intensity is defined. Add to the figure captions which eqs. are employed.

6) It might be useful to present some explicit examples (means explicit values of E and B fields) to illustrate the calculations, e.g. in plasmas or in an accelerator were a B-field is induced due to the movement of charged particles in an electric field.

Author Response

Dear reviewer,

Thank you very much for your valuable comments!
We omitted just one step in derivation of the X-matrix element. So we think that there is no need in the Appendix.

1) To make calculations clear we added equation (28). It makes derivation of the X and Z matrix elements the same

2) The analytical expressions for the Wigner d-functions are very complex. The most common way to write the analytical expression for them is the usage of Jacobi polynomials. However, there is no need in that. This is the main feature of the our paper. General relation for the Wigner d-functions allows us to perform these calculations for ANY transition with n'-n=2!

In various literature, the indices of the d-functions are defined in different ways. In the Varshalovich D. A., Moskalev A. N., Khersonskii V. K. Quantum theory of angular momentum one can find the table with connections of definitions of the d-functions in different books. We used definition from Landau L. D., Lifshitz E. M. Quantum mechanics: non-relativistic theory.

Answer to other comments:

1)Thank you for your advise. We replaced the yellow in the pictures with red. We tried to make the content in the figures as clear as possible. For this, we have added a small clarification about in the description. We encountered some limitations in the latex journal format. Perhaps the editors of the jpurnal will contribute to the size of the pictures become even larger.

2)We want to save the notation for the matrix elements from Gulayev's works. Moreover, these notations are more compact.

3) Using i in this sense is a conventional thing. The original work Demkov, Yu N., B. S. Monozon, and V. Ostrovsky. "Energy levels of a hydrogen atom in crossed electric and magnetic fields." contains this notation. Moreover, in subsequent works about Stark-Zeeman effect the athours use the same notations. For example Novikov, V. G., Vorobev, V. S., Dyachkov, L. G., and Nikiforov, A. F. "Effect of a magnetic field on the radiation emitted by a nonequilibrium hydrogen and deuterium plasma."

4) We have made new figures. So it is ok now.

5)It is very important comment. We have added clarification to the first paragraph of the results section.

6)At this stage of study the Stark-Zeeman effect, we would like to focus on the fundamental quantum features of the problem. We have provided the work with eight examples of calculations. It the future the results of this work will be used in calculations related to the shape of real spectral lines in a magnetized plasma. Hovewer, this is another important problem. In this work, it is important for us to reflect the methodological features of calculations and the general features of spectra in crossed fields.

  I attached revised manuscript to this answer.

Reviewer 2 Report

The paper derives convenient equations for calculations of intensities of radiative transitions between Rydberg states in crossed electric and magnetic fields. The authors should address the following comments.

It is not clear what is new in the present paper as compared to results of Gulyaev (Refs. [10,11]). The authors should clarify this.

The authors should say that Eqs. (2), (3) are possible due to the O(4) symmetry of the hydrogen atom problem and conservation of the Laplace-Runge-Lenz vector. This is the basic physics behind the equations which follow.

The problem of Rydberg states in crossed field was studied by Soloviev and Braun, see Sov. Phys. JETP 58, 63 (1984) and 59, 38 (1984). This work should be cited, particularly in connection with Eq. (6).

i_1 and i_2 in Eq. (7) should be defined.

The first paragraph of Conclusion looks more like an introduction. The authors should rewrite it by focusing more on summarizing the results obtained in the paper. The same is pertinent to the Abstract as the most part of it just introduces the problem.

Some typos and grammar mistakes should be fixed, e.g. H. A. Bethe and E. E. Salpeter in ref. [14].

Author Response

Dear reviewer,

Thank you very much for your valuable comments!

1)Gulyaev derived approximations for the intensity without a magnetic field. Moreover, he gave the solution to this problem in this case. We put emphasis on this in different parts of the article. He didn't take into account the presence of a magnetic field. This information is written in the last paragraph on first page(information about Gulayev's work starts in the begining of the second page)

2) It is very important comment about specific symmetry properties of the Coulomb field. We added a mention of this fact after formula 3

3) Indeed, this article should be included in the bibliography. In this work the authors provide the detailed analysis of specific cases of parallel and perpendicular fields. The reference to this work is now after formula 10 (ref. 18).

4) We defined i_1 and i_2. It is done just after eq.7. "where i_{1,2} are projections of (3) on z". 

5) We have rewritten the first paragraph of the conclusion. We splitted it into two paragraphs and get rid of some information from the introduction. We focused on the obtained results.

We would like to leave the annotation as it is. It seems important to us to emphasize the importance of the problem and show why these specific calculations were made in this work.

 6) We have fixed problem with ref.14

I attached revised manuscript to this answer.

Reviewer 3 Report

Report for “Spectra of a Rydberg atom in crossed electric and magnetic fields” by A Letunov and V. Lisitsa

The authors present an analytical simplification for the case of Rydberg states of Hydrogen beta transitions in electric and magnetic fields. The paper is well written, and the mathematics is straightforward to follow.

I find the results useful and well described. The exact details are outside of my field, so I cannot comment on the originality of the work, however the derivation is interesting and the result useful, so I believe it deserves publication.

I have only minor suggestions:

(1) For some reason, spaces have been ommitted before all parentheses, e.g. :”excited(Rydberg)” is missing a space between excited and (Rydberg.

(2) The X, Z & Y in Figures 1 and 2 are not clearly defined in the text or figure captions.

Otherwise, I believe the paper is ready for publication.

Author Response

Dear reviewer,

Thank you very much for your valuable comments!

1) It was an issue with latex. Now we have fixed that.

2)Thank for this comment! We have made additional clarifications in the first paragraph of the Results section.

Reviewer 4 Report

Spectra of a Rydberg atom in crossed electric and magnetic fields

Andrei Letunov and Valery Lisitsa

Referee’s Report

This is a well written paper dealing with the problem of treating excited Rydberg states in the presence of crossed electric and magnetic fields.   A complete quantum mechanical description of the excited Rydberg states becomes increasingly difficult for the high quantum numbers.  And there are good justifiable reasons to adopt, a semi- classical approach which render the calculations tractable. 

The presentation is clear and well structured.  It is shown that the procedure yields satisfactory results.

So on purely scientific grounds, the referee has no major criticism of the present work.  Of course, it is clear that the topic does not seem to have attracted much attention in recent years. Indeed, there has  been little interest in the topic since the work of Gulyaev (ref 10, 11), Demkov et al (ref 16)  Novikov et al.  (ref. 18), over 40 years ago.   So, this present work is unlikely to attract much attention to-day.   

On the other hand, this paper does draw attention to the use of the semiclassical approach of Gulaev, which makes the calculation feasible.  So, this paper does represent a real contribution to our understanding of the spectra of Rydberg atoms in crossed e.m. fields. 

For that reason, the referee recommends publication, subject to minor grammatical corrections of the text.  

Detailed comment.   The phrase, (lines 9 to 12 ), of the abstract is unclear.  The referee would suggest “The matrix elements involving  the presence of electric and magnetic fields are calculated using a representation closely related to the parabolic quantization on two different axes. “

Author Response

Dear reviewer,

Thank you for your comment!

We have made the proposed correction.
